## Overview Review

Indigenous; mental illness prevention; global mental health; diversity; decolonization

**Corresponding author:**
Md. Omar Faruk;
Email: mfaruk2@lsu.edu

# Addressing epistemic injustice in the mental healthcare of Indigenous people in Bangladesh: Implications for global mental health

Md. Omar Faruk[1,2] 

[1]Department of Clinical Psychology, University of Dhaka, Dhaka, 1000, Bangladesh and [2]Department of Psychology, Louisiana State University, Baton Rouge, LA 70803, USA

## Abstract

Indigenous peoples across the world are at disproportionate risk of mental health problems. Colonial hegemony, cultural infiltration, language loss, land grabbing, limited access to healthcare services, including mental health, and geographical isolation – all in synergy – contribute to the heightened risk of developing mental health problems. Epistemic injustice, apparently unrelated, yet another major determinant – can also contribute to the higher prevalence of mental health problems among Indigenous peoples. Systemic exclusion and marginalization of Indigenous people from the generation, dissemination, and validation of knowledge – the central concept of epistemic injustice – provides an opportunity to reflect on the disproportionate rates of mental health problems. If epistemic injustice is left unaddressed, the impetus for Indigenous peoples to participate in conventional health practices would be greatly impeded. In this article, I present the case of Bangladesh, where the conventional mental healthcare system has historically been ignorant of the inclusion of Indigenous people's perspectives and lived experiences, eventually perpetuating epistemic injustice. Finally, I provide a framework to address epistemic injustices to reform mental healthcare in Bangladesh that can inform a system equipped with equitability, accessibility, cultural sensitivity, human rights, social justice, and collaborative alliance – key tenets of global mental health.

## Impact statement

Indigenous peoples in Bangladesh face higher mental health problems due to systemic exclusion from generating knowledge and accessing resources (e.g., healthcare, education), contributing to epistemic injustices. Both testimonial (disregarding Indigenous perspectives and lived experiences) and hermeneutical (discounting Indigenous epistemologies) injustices perpetuate health disparities and restrict the accessibility of culturally informed mental healthcare. This article offers an Indigenous framework to foster epistemic justices in mental healthcare for Indigenous peoples, including Bangladesh, by bringing reformation of health services incorporating their voices, lived experiences, and Indigenous framework of knowledge. The framework seeks to promote testimonial justice by involving Indigenous peoples in the co-creation of knowledge, valuing their lived experiences as credible, in the pursuit of developing collaborative governance and community-based mental health interventions. The integration of Indigenous practices and efforts to reduce interpretative gaps by promoting epistemic pluralism fosters hermeneutical justices. The Indigenous framework recommends strategies that will ensure epistemic injustices are effectively addressed in the mental healthcare for Indigenous peoples in Bangladesh and beyond.

## Introduction

### Epistemic injustice

Originally coined by philosopher Miranda Fricker (2007), epistemic injustice (EI) generally refers to the unfair discrimination against a person or a group of persons with respect to their ability to know things (Kious et al., 2023). In other words, it is a harmful practice of disregarding the capacity of a person to engage in epistemic practices (e.g., disseminating knowledge to others and making sense of own experiences) (Crichton et al., 2017). Although Fricker's conceptualization of EI initially revolved around the experiences of further marginalization of members in disadvantaged groups in the public and private exchange of information, later it spilled over in several domains to call for social change, including in mental healthcare and research (Carel and Kidd, 2014; Michaels, 2021; Rosen, 2021; Kious et al., 2023; Sakakibara, 2023; Okoroji et al., 2023a; Côté, 2024).

## Epistemic injustices in mental healthcare

It is often argued that the field of psychiatry is susceptible to EIs due to its psychiatric concepts and diagnostic criteria, the roles of institutions, and the dynamics in real clinical settings (Kious et al., 2023; Sakakibara, 2023). For example, it has been argued that the experiences of people with mental health problems have been devalued to produce favorable clinical, academic, and professional knowledge (Rose and Kalathil, 2019a). In addition, psychiatrists and psychologists often attempt to conform to the existing policies that are presumed to be credible within the mental health system (Okoroji et al., 2023b). These structural dynamics and devaluation can be classified into two forms of injustices – testimonial and hermeneutic injustice. *Testimonial injustice* occurs when an individual from a disadvantaged community (e.g., Indigenous peoples) experiences EI due to the prejudicial associations between the community and negative stereotypes (Crichton et al., 2017; Kious et al., 2023). *Hermeneutic injustice*, on the other hand, occurs when the healthcare provider or the overall healthcare system lacks the conceptual resources to comprehend the recipient's knowledgeable reports (Kious et al., 2023; Côté, 2024). The testimonial injustices often stem from epistemic authority held by mental health professionals, including non-Indigenous ones. This epistemic authority tends to discount the lived experiences of those with mental health problems, particularly when the experiences clash with the dominant psychiatric or biomedical narratives (Rose and Kalathil, 2019b). In addition, sanism – a deeply embedded system of discrimination and oppression also marginalizes the knowledge of people with lived experience of mental illness, contributing to testimonial injustice (LeBlanc and Kinsella, 2016). The biomedical model of mental healthcare often fails to recognize the complexities of intersectionality, which considers how multiple identities (e.g., race, gender, and Indigeneity) shape mental health experiences. This oversight can exclude alternative understandings of mental healthcare and Indigenous healing mechanisms, paving the way for hermeneutical injustice by interfering with the ability to make sense of the experiences in line with personal and cultural contexts (Hassall, 2024). Both forms of EIs have mental health implications with evidence suggesting that people experiencing EI experience feelings of humiliation, hopelessness, impaired self-confidence and self-efficacy, sense of belonging, denial of healthcare assistance, mistrust for healthcare providers, and challenges in restoring the damage caused by the condition (Omodan, 2023; Hassall, 2024; Côté, 2024). In addition, it also affects the therapeutic relationship between the client and the mental healthcare provider (Sakakibara, 2023). Therefore, to ensure people with mental health problems particularly those from disadvantaged communities are provided with appropriate care devoid of EI and increase their adherence to the healthcare system, continued worldwide discussions on how to effectively address EI at all levels of healthcare should be made an ethical imperative. Failure to do this means EI contravenes the fundamental ethical principle of care (First, do no harm) (Freeman and Stewart, 2019; Della Croce, 2023). Recent developments have regarded EI as a crucial step to reform the traditional clinical care in mental healthcare services (Carrotte et al., 2021; Groot et al., 2022), with evidence showing that people with mental health problems have endorsed these initiatives (Newbigging and Ridley, 2018).

## Epistemic injustices in Indigenous peoples

Researchers have claimed that people with mental health problems are more vulnerable to EI than those with medical conditions (Crichton et al., 2017; Grim et al., 2019; Harris et al., 2022; Yates et al., 2024) because of the deeply embedded social stigma, negative stereotyping, and systemic inequalities based on race and gender sexuality (Newbigging et al., 2024; Yates et al., 2024). However, this vulnerability may be more experienced in those representing minority groups, particularly Indigenous communities, for example. The dismissal of Indigenous people's voices in healthcare settings may result in misdiagnosis, inappropriate treatment, a sense of separation, and distrust toward the healthcare system. The absence of an Indigenous framework or the use of a predominantly Western framework of psychiatry for conceptualizing Indigenous people's mental health problems reflects the practice of hermeneutical injustice. This also leads to a cycle of exclusion, inadequate mental healthcare, and systemic distrust.

EI among Indigenous peoples has been studied in relation to various systemic issues and services. These include human rights (Tsosie, 2012; Townsend and Townsend, 2021), equity and social justice in health professions (Blanchet Garneau et al., 2021), public health education (Orjinta and Mbah, 2022), colonization and its longstanding influence on educational systems (Adams et al., 2007; Heleta, 2016). In addition, EI has been examined in legal contexts (Altanian and El Kassar, 2021) and feminist perspectives (Johnstone and Lee, 2021). In terms of services, research has explored how empowering Indigenous peoples to run their own services (e.g., child welfare services) (Leckey et al., 2022) can address EI. However, despite these discussions, the implications of both testimonial and hermeneutic injustices in mental healthcare for Indigenous peoples remain significantly unexplored, particularly in Bangladesh.

## Mental health problems in Indigenous peoples: Global perspectives

The number of the world's Indigenous peoples is approximately 476 million, containing merely 6% of the global population (Indigenous Peoples, n.d.). Estimates inform that Indigenous peoples make up about 19% of the extremely poor (Indigenous Peoples, n.d.), and the majority (two-thirds, estimating 260 million) of them reside in Asia with 2000 diverse civilizations and languages (Errico, 2017). While they are collectively called as 'Indigenous peoples', there are other expressions to identify them, especially in Asia, such as 'hill tribes', 'Indigenous nationalities', 'tribal people', 'natives', and 'Mountain people' (Errico, 2017). Indigenous peoples across the world have a shared ancestral connection with their geographical location, including the lands and natural resources which are linked to their identity, culture, livelihoods, and physical and mental well-being. The unique cultural identity, including a variety of languages, often results in distinguished customary entities (e.g., leaders or organizations) separating them from the dominant society or culture (Indigenous Peoples, n.d.). Despite the world witnessing significant economic growth over the last few decades, nevertheless, Indigenous peoples have rarely reaped the benefits. Part of this failure is attributable to the nonchalant attitude of dominant groups, which is often manifested in ignoring Indigenous people's perspectives, experiences, and direct or indirect participation in epistemic practices (testimonial injustices) (Okoroji et al., 2023b). The far-reaching impact of this indifference is taking a heavy toll on the mental health of Indigenous peoples across the world, partly due to epistemic injustice – in relation to the systemic exclusion and marginalization of Indigenous peoples in producing, disseminating, and validating knowledge (hermeneutic injustices). Both testimonial and hermeneutic injustices contribute to the ongoing systemic marginalization, increasing the vulnerability of mental health problems (Levin, 2022).

Indigenous peoples worldwide are at greater risk for mental health problems compared to their non-Indigenous counterparts (Kirmayer and Brass, 2016; Zhang et al., 2025). Research has shown that depression, anxiety, psychological distress, stress, including academic stress, suicide, alcohol and or other substance use, and trauma are the key mental health issues among Indigenous peoples (Grayshield et al., n.d.; Armenta et al., 2016; Tucker et al., 2016; Tucker, Wingate, et al., 2016; Paradies, 2018; Chee et al., 2019; Matheson et al., 2019; Li and Brar, 2022). Of all reported mental health problems, the rate of suicide has been found to be disproportionate (0–187.5 suicide deaths per 100,000 population) among Indigenous peoples as evidenced in a systematic review with 99 studies conducted in 30 countries and territories (Pollock et al., 2018). The review also noted that in some cases the rates were more than 20 times higher among Indigenous peoples. Evidence also showed that the rates of suicide are more pronounced among Indigenous youth (Lehti et al., 2009; Harder et al., 2012). Considering the gravity of suicide among youth, it has been labeled as a 'youth epidemic' (Leenaars, 2006). In addition to suicide, the rate of alcohol use is higher among Indigenous peoples, especially among youth (Stanley et al., 2014). Moreover, poorer levels of well-being and evaluation of own health, self-esteem, and life satisfaction have also been reported (Houkamau et al., 2017). Indigenous peoples also experience physiological complications such as obesity, type 2 diabetes mellitus, cardiovascular diseases, increased blood pressure, asthma, excess body fat, poor sleep, poor oral health, and increased tobacco consumption (Gracey and King, 2009; Huffman and Galloway, 2010; Paradies, 2018; Jamieson et al., 2021). All these consequences can have a synergistic impact on the overall life expectancy. For example, estimates show that Indigenous peoples' life expectancy is up to 20 years lower than those of non-Indigenous peoples worldwide (Indigenous Peoples, n.d.).

### Factors impacting Indigenous peoples' mental health

One of the key factors contributing to the disproportionate mental health repercussions among Indigenous peoples across the world has mostly revolved around the impacts of colonization (Duran and Duran, 1995; Brave Heart and DeBruyn, 1998; Brave Heart, 2003; Whitbeck et al., 2004; Gone and Trimble, 2012; Goodkind et al., 2012; Mitchell, 2019; Tan, 2019; González et al., 2022). Due to colonization, Indigenous peoples experience severe social disadvantages including health disparities (Ferdinand et al., 2020; Zhang et al., 2025), economic disparities (e.g., unemployment and poverty), political disenfranchisement, educational barriers, social and cultural suppression (i.e., a loss of cultural identity), legal as well as property rights issues (i.e., land dispossession) (Kirmayer et al., 2000; Gracey and King, 2009), discriminatory legislation as well as policies to deprive rights, orchestration of genocide (Wolfe, 2006), indiscriminate industrialization (Burns et al., 2022), and mental health consequences ("Indigenous Trauma and Healing", n.d.; Paradies et al., 2015; Urrieta, 2019; Maracle, 2021; Ninomiya et al., 2023). The longstanding effects of colonization also include the devaluation of Indigenous voices and marginalized advocacy for rights, reflecting testimonial injustice. Colonial governments also attempted to assimilate Indigenous peoples into non-Indigenous societies (Kirmayer et al., 2000; Gracey and King, 2009), leading them to witness the dispossessing of traditional and sovereign lands, imposition on settlements and displacement, and casting out cultural practices and languages (Kirmayer et al., 2000; Truth and Reconciliation Commission of Canada, 2015; Orellana et al., 2016; Faruk and Rosenbaum, 2022). Complete

extermination of Indigenous peoples and submissiveness toward the authority of colonizers have also been reported (González et al., 2022). This incessant infiltration or invasion has resulted in serious implications for physical and mental health outcomes that can persist through generations after generations triggering an exposure to intergenerational trauma (Kirmayer et al., 2000; Truth and Reconciliation Commission of Canada, 2015; Browne et al., 2016; McQuaid et al., 2017; Greenwood et al., 2018; Zhang et al., 2025) eventually contributing to a lack of conceptual resources for Indigenous peoples to fully articulate their experiences of mental health within Westernized systems (hermeneutic injustice). Considering the gravity of these repercussions, ensuring universal coverage and healthcare equity for Indigenous peoples has remained a constant appeal (Holder and Corntassel, 2002; Pact, 2016; *The State of the World's Children 2021 | UNICEF*, 2021).

### Mental health problems in Indigenous peoples: Bangladesh perspectives

With approximately 1.65 million people in both mainland and hill tracts, Bangladesh is home to 54 Indigenous communities (Hossen et al., 2023). The majority of them reside in the Chattogram Hill Tracts (CHT) regions in the Southeastern parts of the country. They experience geographical detachment, low levels of income with limited employment opportunities, poverty, inadequate healthcare services, insufficient access to clean water and sanitation, limited access to educational facilities, language loss, and inadequate Infrastructure development (Barkat et al., 2008; Kibria et al., 2015; Rasul, 2015; Faruk and Rosenbaum, 2022). It has been shown that the poverty rate is disproportionately higher in CHT than in any other part of the country (Kibria et al., 2015). In addition, they also experience high conflict rates and indiscriminate land-grabbing practices by settlers (Bangladesh – IWGIA – International Work Group for Indigenous Affairs, n.d.; Barkat et al., 2008; Rasul, 2015). Conflicts, land-grabbing, poverty, inadequate access to healthcare, and the loss of language and cultural identity may result in cumulative trauma contributing to conditions such as PTSD, anxiety, depression, and poor physical health outcomes. For example, evidence suggests that Indigenous peoples in Bangladesh are at higher rates of both physical (e.g., diarrhea and dengue) and mental health problems (e.g., anxiety and depression) (bdnews24.com, n.d.; Faruk et al., 2021). Although Indigenous peoples face these rising rates of health challenges, they have only partial and limited access to healthcare services (bdnews24.com, n.d.), including mental healthcare. Differential understanding about the etiology of both physical and mental illness, lower health literacy, and widespread stigma around physical and mental health may have contributed to the poorer physical and mental health outcomes (bdnews24.com, n.d.; Uddin et al., 2012; Rahman et al., 2021; Faruk and Rosenbaum, 2023; Fenta et al., 2024). The outbreak of the COVID-19 Pandemic has further worsened their mental health outcomes (Faruk et al., 2021). The disregard for Indigenous people's voices is reflected in poverty, restricted access to education and mental healthcare, limited or no opportunity to engage in policymaking and development agendas, and widespread stigma surrounding mental as well as physical health signifying testimonial injustices. On the other hand, cultural infiltration and knowledge marginalization (e.g., continued loss of languages and cultural identity, differential understanding of mental health issues) reflect hermeneutic injustices that reduce Indigenous people's ability to frame their experiences and challenges within the dominant sociocultural discourse.

### Addressing epistemic injustice in mental healthcare: Toward a framework

Concerns may be raised over the extent to which EI is related to these mental health challenges of Indigenous peoples worldwide and in Bangladesh. A plausible explanation of this concern would be that EI is not essentially a risk factor for their mental health challenges; however, undoubtedly a perpetuating factor. For example, continued enforcement of systemic inequalities, such as disregarding culturally sensitive healthcare approaches, may reinforce hermeneutic forms of EI. The persistent or pervasive use of dominant cultural frameworks to conceptualize psychopathology may fail to adequately capture or respect Indigenous perspectives and experiences. As a result, they might struggle to communicate their experiences and needs effectively within the mainstream healthcare system, resulting in misdiagnosis, inappropriate treatments, noncompliance to care, and health inequalities (Indigenous Populations Face Unique Barriers to Accessing Mental Health Help, n.d.; Mental Health Effects of Racism on Indigenous Communities, n.d.; Wylie and McConkey, 2019; Goetz et al., 2023). It is, therefore, crucial to put forward the discussion of EI, particularly by researchers and practitioners, preferably considering a framework when highlighting mental health challenges and risk factors among Indigenous peoples. This is important because the EI framework may capture the underlying mechanisms of why risk factors continue to sustain and contribute to mental health challenges among Indigenous peoples. In addition, an EI framework may reduce testimonial injustices by recognizing Indigenous knowledge and experiences, addressing stigma and biases, and improving trust in healthcare services. The framework may also address hermeneutical injustices by the inclusion of cultural context (e.g., Indigenous language, healing practices), challenging dominant paradigms (i.e., biomedical model of psychopathology), and advocating for a more inclusive approach respecting Indigenous epistemologies. The framework may aid to address structural inequalities by adopting a social justice approach, human rights, and cultural competence in mental healthcare. The collaborative nature that reinforces enhanced participation in shaping mental healthcare services will foster empowerment. Finally, addressing EI through the framework may offer the potential to create more culturally sensitive, equitable, sustainable, and effective mental healthcare systems. Therefore, this paper aims to provide a framework (Figure 1) to address EIs in reducing mental health challenges faced by Indigenous peoples.

The framework places epistemic justice at its core, with actionable steps targeting both testimonial and hermeneutical injustices. Epistemic justice in mental healthcare has been traditionally defined as recognizing people with mental health illness as active agents and collaborators in mental healthcare, utilizing voices or lived experiences and trusting marginalized experiences, offering a model of service delivery that considers service user's values or perspectives, and facilitating dyadic conversations with culturally sensitive ways, and developing new avenues and structures to accommodate alternative forms of knowledge (Johnstone, 2021; Okoroji et al., 2023a; Bortolotti, 2025). The framework described in this article incorporates all major aspects of this broader definition of epistemic justice in relation to the mental healthcare of Indigenous peoples. However, a greater emphasis has been placed on the cultural and social contexts and systemic change to achieve social justice, which is central to epistemic justice (Cohen-Fournier et al., 2021; Côté, 2024). In addition, elements of social justice have also been discussed to reduce structural inequalities (i.e., access to mental healthcare), complementing epistemic justice (i.e., recognition and validation of Indigenous knowledge) (Côté, 2024).

The actionable steps within each component of the framework focus on equitability, accessibility, cultural sensitivity, human rights, social justice, and collaborative alliance informed by several theories and frameworks. For example, *equitability*, *accessibility*, and *social justice* components are central to social justice theory in healthcare (Social Justice and Health, n.d.; Braveman and Gruskin, 2003) whereas *cultural sensitivity* and *collaborative alliance* are based on Multiculturalism and Cultural Competence Theory in healthcare (Herman et al., 2004; Pistole, 2004; Whaley and Davis, 2007), and Community Psychology and Participatory Action Research

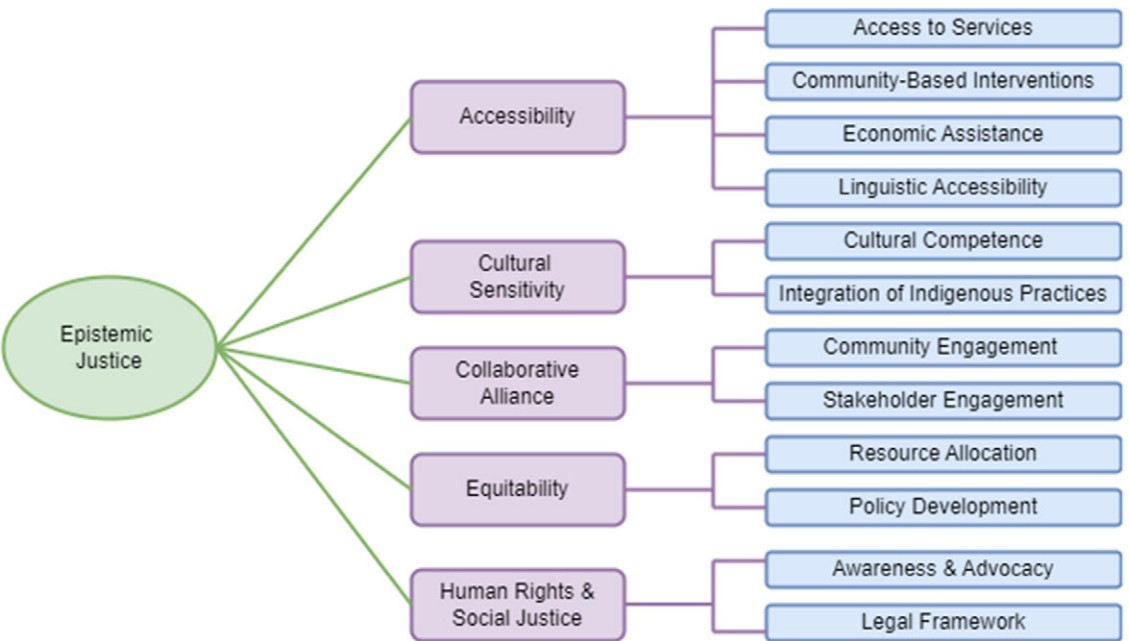

**Figure 1.** The framework to address epistemic injustice in the mental healthcare for Indigenous peoples.

(Lykes, 2017), respectively. Ensuring *equitability* provides an opportunity to address testimonial injustices by ensuring that Indigenous people's voices are heard while fostering *cultural sensitivity* addresses hermeneutical injustices by adopting a conceptual understanding based on respect and shared participation. The *Collaborative alliance* addresses both forms of EIs by fostering the co-creation of solutions and trust. It should be noted that while these action points to address EI were specifically tailored to the Bangladeshi context, the framework could still be applicable and beneficial in other regions.

### Accessibility

To address epistemic injustice in the mental healthcare of Indigenous peoples, increasing access to mental health services is crucial. Research showed that increasing access to mental healthcare is an effective intervention to reduce health disparities (Amaddeo et al., 2001; Nelson, 2002), which will eventually reduce EI, particularly hermeneutical injustice. This can be achieved by focusing on the availability and reachability of mental health services for Indigenous peoples, irrespective of geographical and linguistic differences. This ensures that systemic barriers do not hinder their ability to access necessary healthcare resources. In Bangladesh, however, the availability and reachability of mental health services for Indigenous peoples remains significantly limited due to several systemic barriers contributing to testimonial (i.e., devaluing testimony and biases in the healthcare system) and hermeneutical injustices (i.e., policies that disregard Indigenous epistemologies). For example, the majority of the Indigenous peoples reside in remote hilly and rural areas (Faruk and Rosenbaum, 2023), away from the primary healthcare facilities, which restricts their access to essential mental health services perpetuating testimonial injustices. In addition, there is a pronounced lack of culturally sensitive mental health professionals who are trained to understand and address the unique cultural contexts and needs of Indigenous peoples. It should be noted that the number of mental health professionals in Bangladesh is inadequate and the majority of them provide services in the principal cities (Alam et al., 2021). Furthermore, substantial underrepresentation of mental health professionals representing Indigenous communities remains a major concern (Faruk et al., 2024). It should be noted that there are language barriers which further exacerbate these challenges, as many mental health services are not available in Indigenous languages highlighting hermeneutical injustice. The lack of mental health services in primary languages may result in communication difficulties, eventually impacting trust-building efforts. Additionally, socioeconomic factors such as higher levels of poverty (Kibria et al., 2015) and lower levels of education (Hossen et al., 2023), may impede their ability to seek and receive appropriate care. Discrimination and social stigma surrounding mental health issues within these communities (Faruk and Rosenbaum, 2023) can also contribute to the testimonial injustice leading to the underutilization of available services. Addressing these epistemic injustices require targeted interventions, such as increasing the availability of mental health services, implementing community-based mental health programs, offering economic assistance, and ensuring mental health services are linguistically diverse capturing the needs of Indigenous peoples.

### Service availability

The use of healthcare services for achieving the best possible health outcomes including mental health outcomes is a key to ensuring access to healthcare (Institute of Medicine (US) Committee on Monitoring Access to Personal Health Care Services, 1993). However, due to the prevailing barriers preventing access to healthcare services, many people experience the risk of poor health outcomes and health disparities (Institute of Medicine (US) Committee on Understanding and Eliminating Racial and Ethnic Disparities in Health Care, 2003) which may be attributed to the widespread prejudice or stereotype and failure of understanding the lived experiences of Indigenous peoples in the dominant healthcare framework, contributing to both testimonial and hermeneutical injustices. Efforts to ensure testimonial and hermeneutical justices in the mental healthcare of Indigenous peoples in Bangladesh through increasing service availability may include increasing the number of mental health facilities, mobile health units, and telemedicine.

*Increasing services through mobile units and adoption of digital means.* Currently, mental healthcare is not available in primary, secondary, and peripheral healthcare facilities in Bangladesh (Koly et al., 2022). The services available at the tertiary level are concentrated in urban areas. As a result, these services are largely inaccessible and unknown to those residents in rural areas (WHO, 2016; Hasan et al., 2021; Khatun et al., 2021), especially to Indigenous peoples (Faruk and Rosenbaum, 2023), highlighting structural injustice which is linked to epistemic injustices. In addition to the heavy concentration of mental health services in principal cities, overall mental healthcare delivery is further impacted by the inadequate number of mental health professionals (Alam et al., 2021; Hasan et al., 2021). However, gradual developments in Bangladesh included recruiting psychologists at district and division levels to ensure people with mental health needs have access to mental health services (Faruk, 2022). While this initiative offers a promising solution to the existing scarcity of mental health services across the country, the number of professionals remains a major concern for ensuring mental healthcare for all irrespective of ethnicity with estimates suggesting 0.16 psychiatrists and 0.34 psychologists for 100,000 population (Alam et al., 2021; Hasan et al., 2021). This is equally concerning for Indigenous peoples, particularly those residing in remote hill tract regions who primarily suffer difficulties in transportation. Against this backdrop, establishing more healthcare facilities (e.g., clinics) in these regions would significantly reduce the travel distances for patients with mental health issues, enabling them to seek help. This can be achieved by identifying areas with the highest need through community assessments and ensuring the clinics are staffed with culturally competent professionals capable of understanding the specific mental health issues faced by Indigenous populations. This calls for the use of a systematic approach that may involve epidemiological, qualitative, and comparative methods to understand the mental health problems of Indigenous populations (Wright et al., 1998). This initiative will help ensure testimonial justice by increasing access to services and determining priorities for the most effective use of resources (e.g., implementation of program or intervention) (Conducting Rural Health Research, Needs Assessments, and Program Evaluations Overview – Rural Health Information Hub, n.d.; Wright et al., 1998). Concerted efforts by researchers and practitioners in collaboration with government and non-government organizations may also be useful in ensuring testimonial justice by establishing mental healthcare facilities in accessible areas based on well-thought-out priorities. The first step would be establishing care facilities in urban areas and gradually scaling up the services in remote areas. Consideration of geographical remoteness is important as evidence shows that Indigenous peoples living in urban areas are three times more likely to

receive mental health services compared with those living in regional or remote places (Indigenous Populations Face Unique Barriers to Accessing Mental Health Help, n.d.).

Another way to promote testimonial justice is by increasing the compliance of Indigenous peoples with mental healthcare by making services available in their vicinity. Evidence suggests that compliance can be affected if healthcare facilities are not located within the reach and with distance and transport unavailability also affecting compliance (Indigenous Populations Face Unique Barriers to Accessing Mental Health Help, n.d.; Nolan-Isles et al., 2021; Roberts et al., 2022a). It is reasonable to assume that Indigenous peoples living in the hilly areas in Bangladesh are less likely to avail mental healthcare services not located within reach, due to other pressing needs. Therefore, besides establishing care facilities in the locality, temporary services such as fly-in-fly-out services may be offered in hard-to-reach areas, particularly in the CHT, as recruiting and retaining permanent healthcare professionals is often challenging due to the geographical remoteness (Michiel Oosterveer and Kue Young, 2015). Fly-in-fly-out services refer to a type of service delivery where providers, mostly from the urban areas, travel, stay, and provide services in remote areas and return home for designated periods (Asare et al., 2023; Fruhen et al., 2023). For example, in Australia, fly-in-fly-out services have been in use since the early 2000s for those, including Indigenous peoples, living in remote disadvantaged areas (Smith et al., 2008; Hussain et al., 2015; Nolan-Isles et al., 2021). Other countries, such as Canada, have also introduced fly-in-fly-out services for Indigenous peoples (Roberts et al., 2022b). Despite some difficulties associated with the fly-in-fly-out service model (i.e., insufficient time for consultation and establishing professional relationships, and the risk for vicarious trauma and other negative mental health consequences among service providers), this service has shown promising outcomes in providing mental healthcare in a culturally appropriate manner (Nolan-Isles et al., 2021; Roberts et al., 2022a; Asare et al., 2023; Fruhen et al., 2023). It is important to note that fly-in-fly-out service providers must be capable of cultivating tolerance and respect for inclusion and diversity, which promote hermeneutical justice.

With the advent of technology, health systems worldwide have been capitalizing on the benefits by making significant investments in digital health for health promotion and ensuring access to healthcare (Michie et al., 2017; Dawson et al., 2020). Digital health solutions, often known as 'virtual healthcare' and 'healing at a distance' for mental health and well-being, have also witnessed significant growth recently (Telemedicine, 2010; Marzano et al., 2015). In Canada and Australia, for example, the utilization of digital health solutions such as telehealth to address Indigenous people's mental health problems and gaps in services has been recommended (Province of Manitoba | Mental Health and Addictions, n.d.; Dawson et al., 2020). The use of tele-mental health services offers more accessible and flexible healthcare, reducing stigma, transport expenses promoting autonomy and engagement with services (Chakrabarti, 2015; Hubley et al., 2016). Digital health solutions developed and delivered in a culturally appropriate manner can address hermeneutical injustice by identifying unique mental health challenges experienced by Indigenous peoples (Coleman et al., 2016; Nelson and Wilson, 2017; Hensel et al., 2019; Dawson et al., 2020; Li and Brar, 2022). Telehealth interventions lasting from three to 20 months can be delivered through telephone, internet, and SMS messaging (Dawson et al., 2020). Utilization of telehealth requires enhanced broadband internet access in rural and remote areas across Bangladesh for reliable internet connectivity. User-friendly telemedicine platforms tailored to the needs of Indigenous peoples in the CHT and plainlands, should be accessible on various devices, including smartphones. The Ministry of Chittagong Hill Tracts Affairs has developed an Integrated Digital Service Delivery Platform with a view to providing civic services, including education and training (Ministry of Chittagong Hill Tracts Affairs Set to Digitise All Civil Services by February, 2024). However, currently, the platform does not cover healthcare services within its service provisions. Integration of healthcare services into the platform, including mental healthcare in Indigenous languages provided by trained and culturally competent mental health professionals has immense potential to reduce the burden of mental health problems among Indigenous peoples in Bangladesh. Comprehensive mental healthcare services such as counseling, therapy, support groups, and psychiatric consultations provided through telehealth have shown promising effectiveness (Neufeld et al., 2007; Bashshur et al., 2016; Viswanathan et al., 2020; Di Carlo et al., 2021; Sugarman and Busch, 2023). Despite concerns over data security, technical difficulties such as internet disruptions, efficacy of the intervention being provided, treatment compliance, allocation of resources, handling of emergency cases (e.g., suicidal ideation or attempt), and digital literacy (Hailey et al., 2009; Cowan et al., 2019; Greenhalgh et al., 2020; Siegel et al., 2021; Singla et al., 2022), telemedicine has been a national digital health strategy in many countries across the world (Greenhalgh et al., 2020). To overcome these barriers, including language barriers, interpreter services and infrastructural investments in collaboration with the government and private telecom companies may be introduced. Efforts to utilize low-cost satellite internet services, community Wi-fi hubs, and offline telemedicine applications, particularly in rural areas, may be useful. To protect information, secure storage solutions with end-to-end encryption, training and awareness on data security, and implementing Incident Response Plans to address and tackle data breaches can be enforced. The synergistic implementation of mobile clinics in the form of fly-in-fly-out modalities and telemedicine can improve access to primary healthcare services, including mental healthcare in the rural areas (Gizaw et al., 2022), especially in remote hilly areas. The abovementioned initiatives address both testimonial and hermeneutical injustices by promoting efforts to ensure Indigenous people's voices are heard and respected, which in turn ensures that their experiences are understood and interpreted appropriately.

### Community-based interventions

Amidst the distinct cultural, social, economic, and political characteristics (Indigenous Peoples, n.d.; 10 Things to Know about Indigenous Peoples – United Nations Development Programme | UNDP, n.d.; Subramanian et al., 2006; Bartlett et al., 2007; Faruk et al., 2021), Indigenous peoples including those in Bangladesh have differential ideas of psychopathology (LaFromboise and Fatemi, 2011; Tan, 2019; Faruk et al., 2021). Close connections with the community, people, places, and languages have been integral to Indigenous identities (Maracle, 2021). Therefore, interventions aimed at addressing mental health issues among Indigenous peoples need to be embedded in the community context which promotes hermeneutical justice. Mental health services developed in the Western contexts are seldom sought by Indigenous peoples due to the dominant medical model, which fails to address historical experiences of colonial policies and practices, as well as the importance of shared community perspectives in healing (Lewis and Myhra, 2017; Li and Brar, 2022), perpetuating the cycle of hermeneutical injustices. To promote hermeneutical justice, multi-sectoral and community-based mental healthcare approaches are

critical to addressing structural determinants of mental health and promoting well-being (Castillo et al., 2019; Farah Nasir et al., 2021; Montesanti et al., 2022). Community-based mental health interventions for Indigenous peoples have been postulated as 'culture as treatment' recognizing the importance of integrating traditional practices and community support systems with modern mental health services (Gone, 2013) which aligns with the concept of hermeneutical justices. For example, mental health programs incorporating traditional ceremonies and Indigenous knowledge by elders, reconnection of people with their ancestral lands, culturally grounded indoor and outdoor activities, traditional food gathering, cultural spiritual beliefs, storytelling, and community gatherings were successful in fostering a sense of belonging and cultural continuity, which are essential for mental well-being (Gone, 2013; Farah Nasir et al., 2021; Graham et al., 2021). Additionally, training and employing Indigenous community members as mental health workers can enhance trust and relatability, ensuring that services are delivered in a culturally competent manner (Kirmayer et al., 2009; O'Keefe et al., 2021). Similarly, interventions focusing on building community resilience and social support networks are also crucial to ensure both testimonial and hermeneutic justice in Bangladesh. These may include facilitating peer support groups among Indigenous peoples, community education workshops, and family counseling sessions that address both individual and collective mental health needs within the community. Community ties expressed through numerous sociocultural practices among Indigenous peoples in Bangladesh (e.g., collective agriculture, festivals, and traditional governance structures) can be utilized to implement these initiatives. These services contribute to epistemic justice by offering a space in which the unique cultural and social needs of Indigenous peoples are expressed and accurately interpreted while involving community members in the design and implementation of mental health interventions.

### Economic assistance

Economic assistance within the accessibility of services involves providing financial support and resources to make mental health services more accessible to Indigenous peoples. Currently, many Indigenous peoples worldwide including in Bangladesh have inadequate or no health insurance coverage further contributing to the barriers to healthcare access and health disparities (Institute of Medicine (US) Committee on Understanding and Eliminating Racial and Ethnic Disparities in Health Care, 2003; Call et al., 2014; Koly et al., 2022). Adequate insurance coverage, funding for transportation to and from mental health facilities, community health funding schemes, subsidies for treatment costs, and financial aid for families affected by mental health issues can reduce barriers to accessing mental healthcare and improve well-being (Gizaw et al., 2022). Several countries such as Australia and Canada have allocated funds to foster culturally appropriate mental health services for Indigenous peoples (Canada, 2023; Care, 2024). Reducing the financial burden in accessing mental healthcare for Indigenous peoples in Bangladesh, economic assistance may encourage them, particularly those in the hilly areas in the CHT, to seek help, which contributes to both testimonial and hermeneutical justice by ensuring that their experiences and needs are understood, respected, and addressed. This economic assistance ensures that Indigenous peoples and their families can seek help without economic constraints. In addition, this support can also extend to creating job opportunities within the community, such as training and employing Indigenous mental health workers, which can simultaneously address unemployment while strengthening the community's capacity to manage mental health issues internally.

### Linguistic accessibility

Indigenous languages worldwide, including in Bangladesh, are on the verge of extinction (Khawaja, 2021; Chiblow and Meighan, 2022; Whalen et al., 2022; Faruk and Rosenbaum, 2023). It is reasonable to believe that the language will be lost with the death of speakers leading to the disappearance of linguistic diversity and contributing to epistemic injustice, particularly hermeneutic injustice. For example, this language loss may prevent Indigenous peoples from transmitting traditional knowledge, rituals, and cultural norms. All these losses may have significant implications for the overall well-being with both physical and mental health adversely impacted (Khawaja, 2021; Faruk and Rosenbaum, 2022; Whalen et al., 2022). The continued risk of extinction with no sustainable efforts to preserve and revitalize Indigenous languages is also likely to affect cultural heritage and future generations, contributing to the lower levels of well-being while perpetuating epistemic injustices. Therefore, to promote epistemic justice in the context of Indigenous people's mental healthcare, revitalization of Indigenous languages is crucial as they often experience language barriers in accessing mental healthcare (Li and Brar, 2022). Hermeneutic justice focuses on linguistic accessibility as an approach to revitalizing Indigenous languages. It ensures that mental health services are available in the native languages of Indigenous peoples. This is critically important for effective communication, as language barriers can significantly hinder the understanding and treatment of mental health issues contributing to epistemic injustice (Peled, 2018). Accessibility of service interventions should include the provision of multilinguistic awareness and translation of mental health materials into Indigenous languages to promote a sense of epistemic humility (Peled, 2018). Evidence suggests that educational programs (e.g., Community Linguistic Certificate) and early immersion programs in Indigenous languages embedded in the decolonial approach are likely to increase language fluency, academic achievement, and strengthen community engagement, and offer other social benefits (Khawaja, 2021; McCarty et al., 2021). In addition, state-sponsored activities (e.g., forming a special body to preserve endangered languages) and community involvement can also be helpful in preserving and revitalizing Indigenous languages to protect cultural heritage. Preservation and revitalization efforts in Bangladesh should target achieving epistemic justice by fostering coordinated efforts between Indigenous peoples and expert committee members with substantial linguistic knowledge and understanding. Additional actions might essentially include textbooks in Indigenous languages, the widespread use of technology (e.g., digital recording of languages and creating a website in a particular language), social engagement strategies such as pairing younger people with elderly speakers, the use of archival materials for recreation of languages, and the promotion of values, rituals, and folklores in Indigenous languages in the family and community. Awareness campaigns across Bangladesh highlighting the importance of cultural heritage might help raise awareness among the general population and relevant stakeholders, including government and non-governmental organizations, educators, researchers, and policymakers. Considering these initiatives facilitate both testimonial and hermeneutic justices by reviving and invigorating Indigenous languages deemed critically endangered in Bangladesh. These efforts will increase awareness, foster a sense of belongingness, and help build a practice of passing down the language to future generations. Additionally, incorporating traditional communication

styles and culturally relevant terminologies can enhance therapeutic relationships and improve the outcomes of mental health interventions, contributing to epistemic justice.

### Cultural sensitivity

Cultural diversity is one of the core components of the framework discussed in the current article, as it significantly influences both testimonial and hermeneutical injustices by highlighting the complexities of cross-cultural exchanges, systemic structures, and knowledge frameworks. Cultural diversity influences the overall mental healthcare, including the perception of mental health, health-seeking behavior, and attitudes (Gopalkrishnan, 2018). As culture determines the acceptability of treatment (Hernandez et al., 2009), thus the absence of cultural sensitivity in mental healthcare not only affects treatment compliance but also results in unnecessary and incorrect treatment modalities (Dein, 2018; Fogel et al., 2024) contributing to epistemic injustices. With the distinct cultural characteristics (Indigenous Peoples, n.d.; 10 Things to Know about Indigenous Peoples – United Nations Development Programme | UNDP, n.d.; Subramanian et al., 2006; Bartlett et al., 2007; Faruk et al., 2021), Indigenous peoples worldwide and in Bangladesh have a unique conceptualization of mental health issues (LaFromboise and Fatemi, 2011; Tan, 2019; Faruk et al., 2021), and the failure to consider this uniqueness leads to hermeneutical injustices. Evidence suggests that these injustices resulted in the mistrust in mental health services delivered by non-Indigenous professionals and called for greater consideration of culture (Barron et al., 1999; Schill et al., 2019; Goetz et al., 2023). Therefore, cultural sensitivity remains a key consideration in the diagnosis and treatment of people with mental health issues (Fogel et al., 2024), particularly for Indigenous peoples (Goetz et al., 2023). Addressing cultural sensitivity fosters testimonial justice by validating Indigenous people's voices and perspectives in the diagnosis and treatment of mental health issues and hermeneutical justice by developing an inclusive framework within Indigenous epistemologies. To promote epistemic justice in mental healthcare for Indigenous peoples in Bangladesh, I argue that cultural competence and integration of Indigenous practices are crucial.

### Cultural competence

Cultural competency refers to the ability to recognize and address the diverse cultural perspectives and backgrounds of patients (Stubbe, 2020) which is central to hermeneutical justice. Considering its widespread benefits, cultural competency has become a core requirement for mental health professionals working with culturally diverse groups (Bhui et al., 2007) (e.g., Indigenous peoples). Training for mental health professionals may be useful (Bhui et al., 2007) to increase cultural competence in the mental healthcare for Indigenous peoples worldwide and in Bangladesh. A comprehensive cultural competency training may include cultural awareness training and skill development (Reifels et al., 2018) such as educating professionals about Indigenous histories, cultural nuances, and traditional healing practices which will help develop a lens embedded in Indigenous epistemologies and epistemic humility (Côté, 2024). Additionally, the reformation of mental health education curricula incorporating Indigenous perspectives may help future professionals understand cultural competence from the beginning of their training. Regular evaluation of services, seeking feedback from Indigenous clients, and adapting practices based on this feedback are crucial for ensuring hermeneutical justice.

Engaging Indigenous communities including families in developing and delivering mental health services based on Indigenous values ensures cultural appropriateness and builds trust (Laugharne et al., 2002; Reifels et al., 2018). This can be achieved by integrating Indigenous values into mental health services in Bangladesh countering the marginalization that is often associated with testimonial injustice. In addition, increasing the number of mental health professionals (i.e., counselors) representing Indigenous communities must be a priority to ensure cultural appropriateness in Bangladesh and beyond (Haviland et al., 1983; Williamson et al., 2010; Faruk et al., 2024) further promoting testimonial justice.

### Integration of Indigenous practices

The integration of culturally relevant services and Indigenous practices into mental healthcare promotes epistemic pluralism in which multiple systems of knowledge (i.e., Indigenous epistemologies) are recognized in conjunction with dominant Western frameworks. Evidence suggests that Indigenous people's willingness to use mental health services depends on the degree to which these services are culturally relevant and suitable (Garay et al., 2023), which foster hermeneutical justice by bridging the gap in understanding between dominant healthcare paradigms and Indigenous perspectives. Fostering trust and ensuring their voices and narratives are considered in the mental health service delivery process also facilitates testimonial justice.

Integrating Indigenous practices into mental healthcare in Bangladesh may involve incorporating traditional healing methods, cultural values, and community-based approaches to create a holistic and inclusive care system (Westerman, 2010; Dickerson and Johnson, 2011; McClintock et al., 2016; Sabbioni et al., 2018; Harfield et al., 2024). This can be achieved through utilizing hermeneutical resources by collaborating with Indigenous healers to incorporate traditional rituals, narrative ethics, and narrative therapy, storytelling, ceremonies, and healing practices into mental health treatments (Harfield et al., 2024; Côté, 2024). Additionally, involving Indigenous community members in the development and implementation of mental health programs in Bangladesh can also facilitate hermeneutical justices. Studies have shown that incorporating cultural elements can improve engagement and outcomes in mental healthcare for Indigenous peoples (Gone, 2013; Beaulieu and Reeves, 2022; Harfield et al., 2024) advancing hermeneutical justice. A culturally integrated approach focusing on trust and improving mental health outcomes for Indigenous peoples leads to the development of a sense of 'cultural safety' (Schill et al., 2019; Beaulieu and Reeves, 2022; Harfield et al., 2024) and promotes testimonial justice by ensuring that Indigenous peoples feel respected, understood, and encouraged to share their lived experiences without fear of dismissal or judgment.

### Collaborative alliance

Collaborative alliance in mental healthcare refers to the shared relationship between a mental health professional and an individual with mental health issues. The shared understanding of the development of rapport, goals of treatment, and tasks in treatment is a key element of this collaboration. Research evidence suggests that collaborative alliance is a robust predictor of mental healthcare, such as psychotherapy (Wampold and Flückiger, 2023). Fostering community and stakeholder engagement and partnership may be useful in addressing epistemic injustice in mental healthcare for Indigenous peoples.

### Community engagement

Community engagement generally focuses on actively involving Indigenous communities in all stages of mental health program development, ranging from design to implementation and evaluation (Berry and Crowe, 2009; Russell et al., 2023). Indigenous people's engagement in the development of targeted programs and policies promotes epistemic justice in which mental health service barriers are appropriately addressed, the right to self-determination is promoted (Hurst and Nader, 2006; Ferdinand et al., 2020), and cultural knowledge is incorporated into the decision-making processes. In addition, community engagement results in increased access to healthcare services with enhanced trust (Durey et al., 2016). Amid the benefits of this engagement, many countries have restructured their health policies to address health inequalities experienced by Indigenous peoples. For example, the USA, Chile, Brazil, and New Zealand have established national policies to facilitate engagement and recognize the distinct health needs of Indigenous peoples (Ferdinand et al., 2020; Goforth et al., 2022). Adopting such policies may be useful in combatting epistemic injustices in mental healthcare for Indigenous peoples in Bangladesh. For example, community consultations, focus groups, and participatory action research facilitating open dialog, and mutual learning between mental health professionals and Indigenous community members (Durey et al., 2016; Lin et al., 2020; Goforth et al., 2022) may facilitate epistemic practices where Indigenous peoples are encouraged in the production of knowledge. In addition, community-led programs and partnering with Indigenous peoples-led organizations can also help design and implement mental health services that meet the specific needs of these communities (Isaacs et al., 2010; Durey et al., 2016), contributing to both testimonial and hermeneutical justices. Regular community outreach and education efforts, particularly in remote hilly areas, can raise awareness about mental health issues and available services, encouraging individuals to seek help (Isaacs et al., 2010). However, consideration of age, involvement of families, the use of technology, gender, geographical remoteness, and the use of decolonial approaches (i.e., utilizing Indigenous knowledge systems, promoting self-determination, facilitating collaboration and engagement) are must to ensure that community engagement strategies are appropriate and culturally responsive (Hurst and Nader, 2006; Yellowlees et al., 2008; Durey et al., 2016; Goforth et al., 2022; Russell et al., 2023). By employing these strategies, mental healthcare systems can better engage Indigenous peoples, ensuring that services are rooted in epistemic justice.

### Stakeholder engagement and partnership

Stakeholder engagement and partnership involve forming strategic collaborations with a wide range of stakeholders, including Indigenous leaders, healthcare providers, policymakers, and academic researchers. These partnerships ensure that mental health initiatives are comprehensive and sustainable, leveraging the expertise and resources of various sectors to address complex mental health issues effectively (Dudgeon et al., 2014).

Involving Indigenous peoples in the decision-making processes (Chando et al., 2021; Jull et al., 2023), building partnerships with local health agencies, educational institutions, non-governmental organizations (NGOs), and government bodies (Isaacs et al., 2010), capacity building strategies through continued training, providing resources, and regular communication and feedback (Chino and DeBruyn, 2006) can promote epistemic justice in Bangladesh by enhancing stakeholder engagement. These partnerships can foster interdisciplinary approaches and innovative solutions by combining traditional Indigenous knowledge in Bangladesh with contemporary mental health practices (Wendt and Gone, 2012), enabling a co-creation approach to epistemic justice. In addition, collaboration with stakeholders to advocate for policy changes and increased funding for improving Indigenous people's mental health services is crucial.

### Equitability

Equity in mental health refers to the opportunity for all people, irrespective of race and ethnic identity, to have access to services that promote optimal health (Satcher and Rachel, 2017; National Academies of Sciences et al., 2018). Equity in mental healthcare for Indigenous peoples fosters epistemic justice by adopting a multi-faceted approach that focuses on the fair allocation of resources and the development of inclusive policy. This approach helps to address epistemic injustice by reducing historical and systemic disparities that have long affected Indigenous peoples in Bangladesh, ensuring that they receive the same level of care and opportunities for mental well-being as other groups.

### Resource allocation

It is often argued that the disproportionate distribution of resources (i.e., money, power, and community resources) results in inequalities in healthcare (Social Justice and Health, n.d.; Varcoe et al., 2014; Horrill et al., 2018) that eventually perpetuate epistemic injustice. Therefore, to address inequalities, fair allocation of resources (e.g., assessing Indigenous people's needs and prioritizing funding and support accordingly) to develop a culturally safe mental health service system is necessary. Countries such as Canada have started allocating funds to improve the mental health and well-being of Indigenous peoples (Kazi and Mushtaq, 2022).

Establishing and maintaining healthcare infrastructure for Indigenous peoples residing in remote hilly areas in Bangladesh may be prioritized to reduce barriers to accessing mental healthcare. Increasing mental health professionals representing Indigenous communities (Faruk et al., 2024), adequate funding for existing mental healthcare and research to expand the bandwidth of services (Montesanti et al., 2022), developing manuals for mental health professionals in accessible languages, offering training and ongoing supervision via digital platforms, can be a cost-effective means for fair allocation of resources. It is important to note that enhancing partnerships with multiple stakeholders is crucial to ensure that funding and resource allocation strategies are sustainable and scalable. These initiatives foster testimonial justice by ensuring the validation and amplification of Indigenous voices in shaping healthcare policies and programs, while also addressing hermeneutical injustice by bridging systemic gaps in understanding their mental health needs through the integration of their cultural contexts.

### Policy development

The importance of addressing the unique mental health needs of Indigenous peoples by developing relevant policies has been a consistent urge worldwide (Calma et al., 2017; Jongen et al., 2019; Montesanti et al., 2022; Hobden et al., 2025; Zhang et al., 2025), including in Bangladesh (Faruk et al., 2024). The unique needs and preferences of Indigenous peoples (e.g., integration of culturally appropriate practices) incorporated into an inclusive mental health policy can facilitate epistemic justice in the mental healthcare for Indigenous peoples in Bangladesh. For example, the robust mechanism(s) to include Indigenous leaders and community

members in the policy-making processes (i.e., planning, implementation, and evaluation) enable hermeneutical justice by incorporating Indigenous knowledge systems. In addition, this process also promotes testimonial justice by facilitating trust and equity-focused policies, acknowledging and validating lived experiences, setting measurable goals to improve mental health outcomes, and ensuring equitable distribution of healthcare resources. The policy should include ongoing training programs for mental health professionals in Bangladesh to ensure they are equipped to cultivate epistemic humility. The policy should also focus on legislative support advocating for laws and regulations that protect the rights of Indigenous peoples to equitable healthcare, addressing social determinants (e.g., poverty, education, and housing) (Kirmayer et al., 2014) which further advances hermeneutical justice. Collaborative governance structures promote testimonial justice by facilitating cooperation between government agencies, Indigenous peoples-led organizations, and healthcare providers to support the implementation and oversight of equitable mental health services. Finally, a national policy focusing on valuing Indigenous voices and lived experiences and developing culturally safe mental health resources is crucial (Hutt-MacLeod et al., 2019) to promote both testimonial and hermeneutical justice in the mental healthcare for Indigenous peoples in Bangladesh.

### Human rights and social justice

The concepts of human rights and social justice are important to consider when promoting epistemic justice (Côté, 2024). Human rights have been regarded as power catalysts for change in mental healthcare due to their universal and non-negotiable standards (Porsdam Mann et al., 2016). While the human rights-based approach to mental health primarily focuses on promoting and protecting human rights, placing them at the heart of service provision is another crucial concern (United Nations. Office of the High Commissioner for Human Rights, 2006; Curtice and Exworthy, 2010) as it promotes both testimonial and hermeneutical justice. The subsequent section focuses on strategies such as awareness and advocacy, and robust legal frameworks, that can effectively promote epistemic justice in the mental healthcare for Indigenous peoples in Bangladesh.

### Awareness and advocacy

Researchers have long been arguing for awareness and advocacy as strategies to improve mental healthcare (Hann et al., 2015; Aller et al., 2021). Although mental health advocacy has been pioneered to promote the human rights of people with mental health issues and to reduce stigma and discrimination, it also aims to change structural and attitudinal barriers to acquiring optimal mental health outcomes (Saha, 2021). Therefore, it is often argued that mental health policy should include advocacy and awareness activities, which are essential components of the World Health Organization's mental health policy (Minoletti, 2003; Saha, 2021).

Awareness campaigns and advocacy work may play a crucial role in the promotion of both testimonial and hermeneutical justices in the mental healthcare for Indigenous peoples in Bangladesh. For example, testimonial justice can be achieved by highlighting the unique mental health needs and rights of Indigenous peoples, ensuring that Indigenous perspectives and knowledge are recognized rather than dismissed. In addition, the inclusion of mental health advocacy and awareness components in educational curricula in Bangladesh may also be emphasized to foster hermeneutical

justice. Research showed that taking courses related to mental health awareness and advocacy increases students' mental health knowledge and self-efficacy (Aller et al., 2021). Social initiatives, such as a mental health coalition, can be developed that can foster an advocacy movement. For example, in some low- and middle-income countries, such as Sierra Leone formed a coalition in August 2011 involving persons with lived experiences of mental health issues and their family members, mental health professionals, government, and non-governmental organizations, as well as civil society (Hann et al., 2015). This effort has been effective in promoting mental health in Sierra Leone's national-level policy initiatives and fostering research (Hann et al., 2015). These efforts foster testimonial and hermeneutical justices by dismantling stigma and discrimination and amplifying Indigenous voices and experiences, respectively. Similar efforts should be undertaken to improve Indigenous people's mental health outcomes in Bangladesh. Although several government and non-governmental initiatives in Bangladesh have successfully promoted mental health awareness, reduced stigma, and expanded mental health services nationwide (Faruk, 2022; Jahan et al., 2024); however, there has been minimal effort directed toward raising awareness and advocacy specifically for the mental healthcare of Indigenous peoples. Training people and mental health professionals, educators as well as students, collective efforts (e.g., advocacy movement) led by relevant stakeholders including persons with lived experience of mental health issues and their families, and development of national advocacy and awareness policy are strongly recommended in promoting epistemic justice in the mental healthcare for Indigenous peoples in Bangladesh.

### Legal framework

A legal framework in healthcare is highlighted in the United Nations charter, in which the rights of people with mental health issues to treatment have been underpinned (Hamer et al., 2014). Absence of the legal framework has been identified as a barrier to accessing mental health resources among Indigenous peoples worldwide (Payne et al., 2018), including in Bangladesh. Without a legal framework, there may be a lack of formal recognition and protection for Indigenous knowledge, healing practices, and perspectives, which eventually perpetuates epistemic injustices by excluding them from mental health discourse and policymaking. Indigenous peoples in Bangladesh have limited or no opportunity to engage in the development and delivery of mental health services, partly due to the absence of a legal framework that mobilizes their epistemic practices.

Laws and regulations should be formulated to explicitly guarantee the rights of Indigenous peoples that can help to institutionalize the recognition of Indigenous knowledge and perspectives within mental healthcare systems, advancing hermeneutical justice. Legal provisions can also hold healthcare providers accountable for discriminatory practices and ensure that Indigenous peoples have access to legal recourse if their rights are violated, countering testimonial injustices. This legal recognition also helps to promote hermeneutic justice, where conceptual resources are generated to make sense of their experiences (Medina, 2013). The legal framework of mental healthcare in Bangladesh has witnessed a significant change with the noteworthy legislative step of the Mental Health Act 2018 replacing the century-old Lunacy Act 1912 (Karim and Shaikh, 2021). The law emphasizes the rights of people with mental health issues, including protection from discrimination. However, specific provisions addressing the unique needs of Indigenous peoples are limited, highlighting the need for more focused efforts to ensure equitable access to mental healthcare for these people. To

overcome this gap, cultural competency training and capacity building among mental health professionals, extending mental health services through community-based mental health interventions, particularly in remote hilly areas, continued advocacy for fair allocation of resources, and promoting mental health literacy may be useful within the purview of the existing legal framework to foster epistemic justice.

### Theoretical and clinical implications of the framework

The framework for addressing epistemic injustice in mental health-care for Indigenous peoples in Bangladesh has both theoretical and clinical implications across various dimensions. The framework highlights the theoretical importance of equitable access to mental health services. Theories of social justice and health equity underpin the need for accessible and fair distribution of services, suggesting that disparities in access contribute to broader social inequalities (Creary, 2021). The focus on community-based interventions aligns with community psychology theories, emphasizing empowerment, participatory action, and the social determinants of health (Perkins and Zimmerman, 1995; De Weger et al., 2018; Agner, 2021). These theories suggest that mental health cannot be divorced from the social and cultural contexts in which people live (Gopalkrishnan, 2018; Gonzalez-Guarda et al., 2024). Economic assistance and linguistic accessibility draw on theories of economic justice and cultural competence, indicating that financial barriers and language differences significantly impact the utilization of mental health services (Khawaja, 2021; McCarty et al., 2021; Faruk and Rosenbaum, 2023). In addition, the framework highlights the theoretical stance that culturally tailored interventions are more effective. Theories of cultural humility and competence suggest that understanding and respecting cultural differences improve patient outcomes (Bhui et al., 2007; Lok, 2022). Integration of Indigenous practices supports theories of decolonizing methodologies, arguing for the incorporation of Indigenous knowledge systems into healthcare practices (Bodeker and Kariippanon, 2020; The Importance of Indigenous Knowledge in Healthcare, n.d.; O'Keefe et al., 2021; Beaulieu and Reeves, 2022; Harfield et al., 2024). Collaborative alliance emphasizes the need for community and stakeholder engagement reflecting theories of collaborative governance and participatory healthcare, which advocate for inclusive decision-making processes (Berry and Crowe, 2009; Russell et al., 2023). The framework's focus on policy development aligns with theories of institutional change and legal reform, highlighting the role of policies in shaping healthcare practices and ensuring rights (Calma et al., 2017; Jongen et al., 2019; Montesanti et al., 2022; Hobden et al., 2025; Zhang et al., 2025). Human rights and social justice draw on human rights theories, emphasizing the universal right to health and the need for legal and social frameworks to protect this right (Perehudoff et al., 2019; Nampewo et al., 2022).

Mental health facilities in disadvantaged areas, such as remote hill tract areas, can be enhanced by increasing the number of mental health professionals, particularly from Indigenous communities (Faruk et al., 2024). This would mean lesser barriers to accessing mental healthcare due to hermeneutical injustices. Decentralizing healthcare services aligns with best practices in public health (Sapkota et al., 2023). Community-based interventions can utilize community resources, collaborative alliances, and traditional healing practices to inform culturally sensitive and sustainable mental health services. Equipping mental health and allied professionals with cultural sensitivity and competence training may lead to diagnostic accuracy, eventually leading to treatment compliance. Economic assistance and the promotion of linguistic diversity can remove barriers to accessing care, resulting in higher mental health-care utilization rates and optional health outcomes. Ensuring equitable distribution of resources can address mental health disparities, prioritizing under-resourced areas and populations in resource allocation decisions. Developing and enforcing policies that aim to protect the rights of Indigenous peoples to equitable mental healthcare can institutionalize these practices and ensure sustained improvements in care quality and access. Advocacy and awareness efforts can offer impetus for policy changes and resource allocation that prioritize mental healthcare for Indigenous peoples. These implications help to ensure that epistemic justice is at the heart of mental healthcare for Indigenous peoples in Bangladesh.

### Conclusion

Indigenous peoples across the world, including in Bangladesh, experience unique and higher risks of mental health problems due to a combination of historical and systemic factors such as colonialism and lack of access to healthcare. Epistemic injustice plays a crucial role in this issue by perpetuating the marginalization of Indigenous voices in knowledge creation and healthcare, which further contributes to mental health disparities. If left unaddressed, epistemic injustices will continue to pose barriers to accessing and benefiting from mental health services. The framework proposed in this paper will help reform the mental healthcare system in Bangladesh, incorporating principles of equitability, accessibility, cultural sensitivity, human rights, and collaborative alliance. This approach aims to address the epistemic injustices by highlighting the pressing mental health needs of Indigenous peoples and challenging the systemic exclusion that underpins these disparities. It is expected that the integration of Indigenous perspectives and experiences into mental healthcare will enable the framework to ensure effective health services for Indigenous peoples in Bangladesh, with broader implications for global mental health.

**Open peer review.** To view the open peer review materials for this article, please visit http://doi.org/10.1017/gmh.2025.10008.

**Data availability statement.** This work contains no primary data.

**Acknowledgment.** I express gratitude toward the reviewers for their comments to improve the quality of this article. I also express my sincere gratitude to Dr. Sudipta Sarker for his support and encouragement in preparing the manuscript.

**Author contribution.** M.O.F. is the sole contributor to this work.

**Financial support.** The work received no financial assistance.

**Competing interest.** I declare no potential conflicts of interest.

**Ethics statement.** The study did not involve primary data; therefore, ethical approval was not necessary.

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
