## [Reviewer Report]

Review:

Addressing epistemic injustice in the mental health care of Indigenous people in Bangladesh: Implications for global mental health

Thank you for the opportunity to review this very interesting article discussing a framework to address epistemic injustices in mental health care for Indigenous people in Bangladesh. The framework suggested draws on several theories and other frameworks, including in its core concepts like equitability, accessibility, cultural sensitivity, human rights, social justice, and collaborative alliance. It is based on a large body of literature.

This framework has the potential to be applied to various national contexts where indigenous people face similar challenges in accessing quality mental health care. It is a significative contribution to the literature and has the potential to foster change in healthcare services globally. However, I see two main aspects of this article that would require significant changes, that is, the introduction could be restructured and the place of epistemic injustice/epistemic justice in the proposed framework could be clearer.

I would therefore recommend accepting this article with major revisions.

Introduction structure:

The introduction would benefit from a clearer structure.

I would suggest a structure including subtitles. Here is a suggestion of what it could look like based on how the introduction is currently constructed:

1) Epistemic injustice

a. Definition of epistemic injustice

b. Epistemic injustices in mental health care (including possible testimonial and hermeneutical injustices)

c. Epistemic injustices in indigenous people (including possible testimonial and hermeneutical injustices)

d. The lack of research about epistemic injustice in mental health care for indigenous people

2) Mental health

a. Mental health problems for indigenous people globally

b. Factors impacting indigenous people’s mental health

c. Mental health for indigenous people in Bangladesh specifically

3) Epistemic injustice and mental health

a. The need for a framework to address epistemic injustices in indigenous people’s mental health in Bangladesh.

Regarding the suggested point 1 (b and c), it would be interesting to give more weight to both testimonial and hermeneutical injustices that may be experienced by people living with mental health problems, indigenous people, and indigenous people living with mental health problems (e.g., epistemic authority held by mental health professionals and non-indigenous professionals, the opposition between a biomedical model of mental health and a maybe more holistic vision of healing in indigenous communities, different practices, etc.). Maybe the intersectionality framework could provide important foundations to integrate this dual oppression of being indigenous and living with mental-health problems. For example, regarding hermeneutical injustices, it would be interesting at p. 3 to have a bit more background on how indigenous knowledge may be marginalized, resulting in hermeneutical injustice. Sanism could also be an interesting concept to integrate.

Regarding point 2, I would suggest this structure to avoid considering indifferently all indigenous identities. Indeed, indigenous people may have quite different life conditions, contexts and history depending on the country. Indigenous people from North America, South America, Asia, Africa and Oceania may differ substantially. The cited literature include literature on Indigenous’ mental health in Alaska, Australia, Canada, United States, etc. where policies, healthcare structures, beliefs, etc. may be substantially different, thus influencing the care provided, the needs, the challenges, etc. Therefore, presenting first how indigenous people are globally impacted by mental health problems, the factors leading to these problems and then talking specifically about the context in Bangladesh would allow for a clearer structure and would avoid going back and forth between a global and a local context. Regarding this point, I would also suggest mentioning clearly why indigenous people from several location are discussed together (e.g., some common or similar factors may impact their mental health in similar ways, as they share similar challenges).

Regarding point 3, the article already clarifies well why such a framework is needed. I would only recommend structuring the argument so that the reader easily understands how it could be beneficial to reduce both testimonial and hermeneutical epistemic injustices for indigenous people living with mental health problems. The place of epistemic (in)justice in the framework could also be further discussed.

The framework

In the different sections describing the specific components of the proposed framework, little to no place is made for epistemic injustices and epistemic justice. I understand that this model aims at a better “epistemic justice” in the mental health care of indigenous people in Bangladesh, however, this is not reflected in the description of the components.

First, as epistemic justice is at the core of the framework, it would be useful to provide a clear definition of that concept. Is it equivalent to social justice that you use as a concept across the text or is it a different concept? Is epistemic justice involve simply not committing epistemic injustices or does it imply something else, such as anti-oppressive approaches?

Second, in the different sections of the article explaining the components of the framework, clear links with the concepts of epistemic (in)justice are needed. In the current state of the article, it seems like this model promote social justice, health equity, etc., but not really epistemic justice. Clear explanations on how each component is anchored in epistemic justice is needed. Otherwise, I would recommend not talking about epistemic (in)justice at all in the article and using a framework of health equity or social justice. How are health inequities based on epistemic injustices? How epistemic justice is being promoted by the components suggested in this framework? The article should answer these questions clearly. For example, for the section on “Linguistic Accessibility”, I would recommend this article by Yael Peled (2018) to integrate epistemic (in)justice in the framework:

Peled, Yael. 2018. Language barriers and epistemic injustice in healthcare settings. Bioethics 32 (6): 360–367. https://doi.org/10.1111/bioe.12435.

Regarding your section “Integration of indigenous practices”, I would recommend works on Epistemic Pluralism and discussing hermeneutical injustices. I briefly covered this point in my article that you cite earlier in your text:

https://link.springer.com/article/10.1007/s11019-024-10210-1

And you can also refer to the following text that is specifically on indigenous context (in Canada):

Cohen-Fournier, Sara Marie, Gregory Brass, and Laurence J. Kirmayer. 2021. Decolonizing health care: Challenges of cultural and epistemic pluralism in medical decision-making with indigenous communities. Bioethics 35 (8): 767–778. https://doi.org/10.1111/bioe.12946.

In the same vein, the figure representing the framework is well-organized and clear, but it would be interesting to see how each component is related to testimonial, hermeneutical or both types of injustices and to explain briefly in the description of the model how they relate to each other.

I strongly encourage the author to submit a revised version of their work. The framework suggested is interesting and has great potential. Better integrating the concepts of epistemic (in)justice would increase the overall quality of the model and of the article and would provide a concrete theoretical model to promote epistemic justice in mental health care for indigenous people, that could also be applied to other health conditions or national contexts. Thank you again for the opportunity to review this article.

---

## [Reviewer Report]

Since this paper is based on the needs of Indidgenous Bangledeshi, more focus should be placed on their contexts and needs. Although all the appropriate people are quoted, what is suggested could apply to any vulnerable group (probably because so much has been written about so many) and the point, therefore, needs to be made about the particulars of this group so that the contribution of the author is truly seen as original and teaching the rest of us things we don’t already know,.Since there is a fair amount of repetition in this paper, some of that can be replaced by presenting more information about this group (ex. history, social conditions which fuel the EI, privileges of the dominant group, caste issues to overcome, transmission of intergenerational trauma etc.)

There are many themes and contexts within this paper. Perhaps, the author should consider dividing this paper into two articles: one on the issues and particularities of the Indigenous people of Bangladesh and why these would be best served throught a recognition of the EIs and why that is important and another on modalities which may be useful in addressing the issues raised.

There are also technical grammar issues such as having noun and verbs agree and inserting indefinite articles when necessary (often occurs because the author is so familiar with the material that these slight infractions are not readily noticeable). This could be easily solved by having a disinterested party read and suggest corrections before suhbmission. This will not be too time-consuming.

I have highlighted some specifics which I hope will assist the author(s). Since a page of instructions to reviewers was placed at the beginning of the manyscript. The page numbers used in this review may be one lower that the authors' (ex. my page 2 may be your page 1).

P 2 (Abstract)

line 6 - suggest ‘peoples’ indicating that it is understood that indigenous people are not a monolithic group. Should be addressed throughout the paper.

P4

line 32. this is merely a suggestion but I am concerned that those who do not understand epistemic injustice or seek to devalue it, may construe “inflated degree” to mean that the experience is being ‘exaggerated’. Should a less interpretative word be used?

P 5

line 4: not sure what is meant by ‘sufficient’; how is that measured and is it a level which must be attained by all or is it limited to what can be expected from a particular individual, given their level of exposure to “others”?

IS THE EFFICIENCY ATTAINED THROUGH SPECIFIC TRAINING?

line 13? Why is it ‘unfortunate’? and why the emphasis on ‘suggested’? the very nature of psychiatry (and the other helping professions) makes it susceptible to epistemic injustice.

line 24 -26 should be ‘to shape’ or ‘in shaping’. Also, there is something missing in the sentence “in addition...2023.” as it is not clear what is being stated.

lines 25 -33. Suggest separating what the individual may feel from what they may have to endure from the system as currently they run together and it is difficult to determine what issues belong to each category.

lines 43 - 44. ‘continued world-wide discussions on how to’ might be easier for the reader to follow.

line 44: I think the author means ‘failure to do this’

line 48 -54: this sentence seems to contradict the preceding one. Removing ‘However’ could solve the problem if the contradiction is not intended.

P6

line 9: suggest ‘educational systems’

line 34: the world’s

lines 41 - 47 It is not clear whether this refers to all indignenous peoples or only indigenous peoples in Asia. Also, ‘ethnic minorities’ in the Global North is not reserved for indigenous persons. In fact, it often does not refer to that group at all but to non-indigenous subgroups within the dominant population. Also, ‘as’ before indigenous should be removed.

P 5

line 6: ‘participation’ in what?

line 21: does ‘contained’ in this instance connote the same as ‘are’?

line 26: does ‘amongst them’ refer to the variables just mentioned, such that having these things lead to suicide? or does ‘them’ refer to indigenous peoples, especially since indigenous ‘people’ are mentioned later in the same sentence?

line 28: should be ‘as evidenced’ or as is evident.

line 58: unclear what is meant by ‘plain’

P8

line 16: who is engaging in ‘indiscriminate land-grabbing practices’?

lines 20 -26 It is not clear how the issues mentioned in line 16 relate to the outcomes in lines 20 - 51. some elucidation is necessary. also should be ‘these growing prevalences’ or ‘this growing prevalence’. Furthermore, stating that ‘despite’ the prevalances, Indignenous people have ‘limited’ access, reads as though ‘limited’ is positive. perhaps, ‘some access, though limited’?

There are many valid arguments contained within this paragraph. However, they tend to run into each other. Perhaps, dividing the paragraph into segments would increase clarity and comprehension for the reader.

P 10

line 10. it is, perhaps, not a good idea to a priory offend readers who may disagree with your argument by referring to them as ‘naive’. Moreover, is the argument being made that although some would say it is not a risk factor, they would “undoubtedly” say that it is a perpetuating factor? suggest clarification is needed.

lines 14 -17 it is not clear what is being said in that statement. Perhaps, it needs rephrasing?

line 35: put forward by whom?

P 10 - 11:

lines 43 - 18

Suggest explaining what the framwork is about before before discussing it. The figure could be introduced when the framework is first mentioned which would make the arguments easier to follow.

p10

lines 32 -60, suggest italicizing the elements of the framework in the narrative so that they become distinguishable. ex.Cultrual sensitivity.

p 12

Lines 29 - 32 somewhere the reader needs to be told what is the unique contexts of Bangladeshi indigenous people

lines 40 - 60 surely it goes beyond language, do such things as caste also play a role? what are some of the factors which may promote Indigenous underutilization of availabel services?

Although what is written is important, it is difficult to understand how they differ from any other vulnerable community.

P14

lines 43 -44 unsure what “health needs avail the needed care” means

line 52, ‘respectively’ is confusing as the proportion of psychiatrists and psychologists have already been made clear.

p 15

lines lines 33 -45. Is is being argued that the closer the services are located, the more likely folk are to use them? if so, suggest making it clearer.

lines 52 - 56 does this mean they stay at home for a designated period or in the Indigenous community?

line 57: tell the reader how long this has been used in Australia. It is also used in other places - for example, Canada

P 16

line 45 ‘than’ is not clear in this sentence

p 18

lines 20 - 21 unclear what “being on land” means

line 42 should this be ‘while involving’?

p 19

line 20 either communities' capacity or the community’s capacity

line 41 is there a clearer way to say “contributing to the compromised well-being”? As it is, it could be interpreted that contributing to compromised well-being is a good thing.

P24

line 5: what are some of the decolonial approaches; what does “are must to ensure that community” mean?

P 25

Lines 25 - 28

How will the “fair allocation” be done? I assume that this should involve the Indigenous people in the deliberation?

P 26

Line 5: Robust mechanism(s). Please go through the document and fix error such as this..

line 9: what is meant by trust equity-focused policies?

lines 38 -40: these statements need some elucidation

lines 43 -49 unclear what is being said here especially "yielding their synergistic benefits’s

Lines 53 - 56: is this sentence complete? seems to be missing something

Lines 56 to P 27, line 3. This bit is unclear.

In conclusion, this paper is more focussed on Indigenous persons in general than Bangladesh in specific. Upon To address this issue, the author first has to decide on what they would like the reader to focus and proceed from there. Since many authorities have addressed the framework being proposed - albeit not in the manner in which they are presented in this paper, it would have been interesting to read how they pertain to Bangladesh in particular.

---

## [Editor Report]

Dear authors,

Please find the reviewers' comments below. Both reviewers felt that your manuscript is well-written and appealing, and that the proposed framework is potentially useful for examining epistemic injustice in the Global South. However, they had major critiques regarding the nature and structure of some of the sections, which will require major revisions. Please prepare a detailed point-by-point response letter and revise your paper accordingly.

Best,

Franco

---

## [Reviewer Report]

This article should be published but it needs to be edited, both for length and clarity. In the beginning just let the reader know that when you say worldwide, you are including Bangladesh and that the article is focussing primarily on Bangladesh. then you don’t have to keep saying ‘worldwide including Bangladesh’ as it would be already understood.

Including Bangladesh should only be rotated if there is a possibility of confusion to the reader. There is also a need to construct sentences so that readers are clear when the end has been reached. This can usually be done through the use of a conjunction, such as ‘and’ after the serializations. Suggest a reduction of repetition. For example, we are told on many occasions that the dominant culture is not being receptive to the voices and lived reality of indigenous peoples As important as that is, stating it in practically every paragraph runs the risk of losing the reader and the author’s very important message being loss or ignored. I suggest a deep edit.

P 1

29 -32 I understand what is being said. However, it still reads as though the authour is proffering that the best thing that can happen is for indigenous peopoles to “adhere” to “conventional health” norms. I posit that it would be more inclusive to state that if epistemic injustice is not addressed, the impetous for indigenous persons to participate in conventional health practices would be greatly impeded. I also do not understand the usage of the word “despite” in this context as it seems to be another indication of the presumed supremacy of 'the conventional health" system.

line 36, is the conventional system ignorant of the inclusion or the exclusion?

P2

line 6: suggest ‘including’ Bangladesh rather than “and” (more inclusivity).

line 41 either ‘an indigfenous framework’ or ‘indigenous frameworks’

P4

line 60: suggest replacing second “professionals” with ones.

P 5

lines 13 - 15. This is a bit confusing as intersectionality has to do with how the myriad identities which we all carry contribute to the interpretation of events

line 22: perhaps replace the second “experience” with somethinglike ‘undergo’

line 40: should “regraded” be ‘regarded’?

P6

lines 17 - 32. it is unclear which of these are services and whci are issues to be addressed. Suggesting fixing it for clarity.

line 53 ‘and’ physical as well as mental well-being. Suggests reviewing the manuscripts for these types of omissions as there are a few and they interfere with the flow.

P 7

line 3: nonchalant not nonchalance

Lines 9 -18: are there any references to strengthen this observation?

lines 21- 26: perhaps leave it at ‘substance abuse’ and remove marijuana as there is much controversy concerning this substance and it is often prescribed in the health systems of dominant societies to assist patients.

Also, citations which are placed within parentheses should be using & and not and. Thus, (Kirmayer & Brass)

P8

line 43: and casting out

P 9

line 16 “too” is not needed as is reads as though Bangladesh has been excluded up to this point.

p 10

line 47: hermeneutic forms or a hermeneutic form

line 49: framworks the dominant framework

P11

line 31: this paper aims is more accurate as it is currently being written

line 35 Perhaps this would be a good place to insert the Framework so that readers can follow it as they read. Two pages down is quite a bit away, as the Frameork has now been discussed for a while.

P 22

line 18: I suggest the author perform a check and reduce the instances of using the same word in the same sentence, whenever possible. It can be very distracting.

P 23

line 34: suggest inserting something like ‘of indigenous peopes’ efter “experiences”. This is for clarity.

P 26

line 53: is the argument that interventions need to be embedded in a community which pormotes hermeneutics or that such embedment would support hermeneutics? If the latter, the insertion of ‘then’ or ‘in turn’ after “which” should make it clear.

line 60 is the author saying that the inclusion of “shared community perspectives” perpetuate the cycle of injustices?

P28

line 60: suggest inserting ‘many’ after currently. For example, Indigenous peoples in Canada have access to health insurance and health care. There is, however, often racism and other barriers in the manner in which is delivered - but it does exists.

P 31

line 31: the current article?

P34

line 51: do you mean ‘leads to developing" or ’the development of' ?

P38

line 15: not sure what is being said from “while...contexts”

---

## [Reviewer Report]

Thank you for the opportunity to review this manuscript a second time. The authors did a really great job integrating the comments reviewers made to their last version of this manuscript in an improved manuscript that is clearer and better structured.

My main comment for this version would be that the article is quite long in its actual form. I would suggest shortening sections when possible and make the arguments more straight-forward.

---

## [Editor Report]

Dear Authors,

The reviewers found this version of the manuscript compelling and worthy of publication. However, they expressed concerns regarding its length and a lack of clarity in some sections. Please revise the manuscript once more and send it back at your earliest convenience.

Thank you.

Best, Franco